# Novel Biomarkers and Druggable Targets in Advanced Melanoma

**DOI:** 10.3390/cancers14010081

**Published:** 2021-12-24

**Authors:** Pier Francesco Ferrucci, Emilia Cocorocchio

**Affiliations:** 1Cancer Biotherapy Unit, Department of Experimental Oncology, European Institute of Oncology IRCCS, 20141 Milan, Italy; 2Department of Hemato-Oncology, European Institute of Oncology IRCCS, 20141 Milan, Italy; emilia.cocorocchio@ieo.it

**Keywords:** biomarker, melanoma, molecular targets, immunotherapy, liquid biopsy, microbiome

## Abstract

**Simple Summary:**

Immunotherapy and targeted therapy represent effective therapeutic opportunities that radically changed the available armamentarium for the treatment of melanoma. In about 50% of patients with advanced disease, long-term survival can be achieved; unfortunately, the other half of patients still have limited benefit from such innovative therapies and experience complications that affect their quality of life. Affordable, reliable and easily-to-detect biomarkers are urgently needed to facilitate the decision-making process, in order to identify patients best suited to receive immune or targeted therapy, with the aims of reducing toxicities, enhancing efficacy and preventing recurrences.

**Abstract:**

Immunotherapy with Ipilimumab or antibodies against programmed death (ligand) 1 (anti-PD1/PDL1), targeted therapies with BRAF-inhibitors (anti-BRAF) and their combinations significantly changed melanoma treatment options in both primary, adjuvant and metastatic setting, allowing for a cure, or at least long-term survival, in most patients. However, up to 50% of those with advance or metastatic disease still have no significant benefit from such innovative therapies, and clinicians are not able to discriminate in advance neither who is going to respond and for how long nor who is going to develop collateral effects and which ones. However, druggable targets, as well as affordable and reliable biomarkers are needed to personalize resources at a single-patient level. In this manuscript, different molecules, genes, cells, pathways and even combinatorial algorithms or scores are included in four biomarker chapters (molecular, immunological, peripheral and gut microbiota) and reviewed in order to evaluate their role in indicating a patient’s possible response to treatment or development of toxicities.

## 1. Introduction

Ipilimumab was the first immune checkpoint inhibitor to show improvement in overall survival (OS), with a benefit that could last years after drug discontinuation. Unfortunately, its low and slow kinetics of response combined with the frequency of drug-related toxicities limited its application. Anti-PD1/PDL1 and anti-BRAF treatments are better tolerated and have a higher efficacy, but some patients still develop peculiar complications, which are difficult to predict or prevent. Furthermore, several immune combinations (anti-CTLA4/anti-PD1s, anti-LAG3/anti-PD1, NKTR/anti-PD1) or even immune and target combinations (antiPD1/target-therapy) are under investigation, with some efficacy advantages, but also frequent and severe toxicities, so that they cannot be offered to every patient.

For all these reasons, affordable and reliable biomarkers are urgently needed to help with identifying the best patients for specific treatments. In particular, early recognition and intervention on toxicities is critical, as many patients require treatment interruption, discontinuation and/or long-time immunosuppression that significantly affect their quality of life and response. Several biomarkers (molecular, immunological, cellular and humoral) have been investigated in order to identify those patients who could expect the best outcome from treatments and those at risk of developing severe toxicity.

In general, biomarkers can be prognostic or predictive: the former are able to give information about the outcome of disease independently from intervention, and the latter allow for stratification of patients with respect to benefit from a given therapy. However, some biomarkers retain both a prognostic and predictive role. Predictive biomarkers may help including patients within a specific category, such as for the mutational status of a given gene (BRAF in melanoma, cERB-B2 in breast or EGFR in non-small-cell lung cancers), or indicate a continuous variable, such as the PDL-1 staining percentage. In this case, they are often dynamic, and the arbitrary cut-offs to determine ‘positivity’ may incorrectly classify whether the patient will or will not benefit from the specific intervention. To be reliable, biomarkers should also be independent from other routinely used predictors, and subjected to continuous revision, as our biological understanding of the disease improves over time. On the other hand, due to the complex interaction between host and disease, building a sort of algorithm or signature, which takes into consideration different biomarkers at the same time, rather than one single factor, would increase their clinical value in terms of reproducibility, specificity and sensitivity.

Hereon, we will review the role of the principal established and innovative biomarkers, analyzed as single molecules or in combination, to define the best algorithms for melanoma management.

## 2. Molecular Biomarkers

### 2.1. BRAF

About 50% of melanomas harbor oncogene BRAF mutations [1] able to activate the MAPK pathway, which regulates a melanoma cell’s proliferation, growth and survival. The most frequent mutation on the BRAF oncogene is V600E, while V600K, V600R, V600M and V600D occur more infrequently [2].

BRAF V600 mutations are usually associated with a more aggressive disease and a shorter survival in stage IIIB, IIIC radically resected and even stage IV melanoma, potentially arguing for their prognostic role [3,4].

On the other hand, BRAF V600 mutation is a well-defined predictive factor for a novel class of drugs specifically designed to inhibit BRAF (also known as target therapy), resulting in amelioration of prognosis [5].

Testing of BRAF V600 mutation is now recommended in stage III and IV melanoma, since the presence of the mutation represents an ideal target, allowing the use of BRAF-inhibitors (BRAF-i), in both the adjuvant and the metastatic setting. BRAF-i selectively binds to and inhibits the activity of the gene by blocking its constitutive activation. Initially, BRAF-i were used as monotherapy, allowing dramatic response in terms of reduction of tumor size. However, BRAF-i were burdened by low duration of response and cutaneous toxicity with secondary tumors development (mainly skin cancers) [6,7]. Moreover, paradoxical activation of the MAPK/ERK pathway through BRAF amplification or alternative splicing, loss of PTEN, COT overexpression, RAC1 alteration, MEK 1/2 mutations and NRAS mutations are recognized mechanisms responsible of primary/acquired resistance or secondary tumors occurrence. The addition of the inhibition of the downstream effector MEK improved progression-free survival (PFS) from 6.2–8.8 months to 9.9–11.4 months compared to BRAF-i alone [8,9], also reducing the secondary skin cancers rate (7% vs. 19%) [10]. These data support the BRAF-i/MEK-i combination as the standard treatment in BRAF-mutated melanoma.

Another relevant aspect is the heterogeneity of BRAF mutational status. In advanced disease, discordance between primary and metastatic lesions was reported in at least 40% of lesions [11,12]. BRAF mutation heterogeneity seems to increase with increasing numbers of metastasis [13], reflecting the need for testing in the most recent ones. Furthermore, BRAF mutation accounts for the differential response of metastatic lesions and disease progression during target-therapy. BRAF mutations and BRAF-i are also influencing the tumor microenvironment: BRAF V600E mutation promotes stromal-cell-mediated immunosuppression via IL-1 induction [14]. On the other hand, BRAF-i can increase melanoma antigen presentation and T cell infiltration and reduce immunosuppressive cytokines. In primary setting, a particular immune signature (i.e., CD8+T cells genes) may be predictive of BRAF-i/MEK-i activity [15]. Some immune-escape mechanisms of resistance to BRAF-i were suspected by identification of markers of immunosuppression (TIM-3, PD-1 and PD-L1 expression). Although the reasons of BRAF-i resistance are numerous and multiple, and sometimes molecularly very distant, this information provides the basis for the combination of immunotherapy and target-therapy in BRAF-mutated melanoma [16].

### 2.2. NRAS

About 20–30% of melanomas harbor NRAS mutations, which are also able to activate the MAPK pathway. The most frequent NRAS mutation is located in exon 1 (codon 12) and exon 2 (codon 61). NRAS mutant melanomas seem associated with worse prognosis [17] and with an increased risk of brain metastases. Currently, no target agents against NRAS mutation are available, and MEK-I, such as Binimetinib, demonstrate modest activity in this subset of patients [18]. Although NRAS mutation is often associated with an increased mutational tumor burden, with a potential benefit from treatment with a checkpoint-inhibitor, Kirchberger et al. reported similar result for checkpoint inhibitors in NRAS-mutant and WT patients, with a reduced survival in NRAS-mutant ones [19]. Nowadays, detection of NRAS mutation does not have an established predictive role, while it could add prognostic information in melanoma patient (for example, increased risk of brain metastasis).

### 2.3. KIT

KIT is a tyrosine-kinase receptor that plays a role in the development of numerous cell lineages, including melanocytes, mast cells and hematopoietic progenitor cells. It is able to activate downstream effectors through MAPK and AKT pathways.

KIT alterations include amplifications or mutations, which are rare in cutaneous melanoma (about 3%), but relatively frequent in mucosal, acral and cutaneous melanoma with chronic sun damages (9–21% of cases) [20]. In several phase II trials, Imatinib proved to be effective in patients with metastatic melanoma harbouring KIT point mutations mainly located on exon 11 and 13 [21]. However, toxicity is significant and treatment continuity or dosing can be complicated, influencing effectiveness.

Other selective KIT-inhibitors (with lower IC50) that demonstrated some kind of activity in KIT-mutated melanoma are nilotinib [22], avapritinib [23] and masitinib (NCT01280565). In the masitinib trial, which was halted due to slow accrual and burocratic issues, ORR was 39.1% (*p*-value = 0.0012) in first line of treatment and 33.3% (*p*-value = 0.0049) regardless of the lines of treatment. There were also two complete responses, including one complete response lasting for 1030 days.

Currently, it is recommended to test KIT mutations in mucosal, acral or primary unknown melanoma, while it should be considered in cutaneous melanoma with chronic sun damage. However, its prognostic and predictive relevance should still be established.

### 2.4. PTEN

Phosphatase and tensin homolog deleted on chromosome 10 (PTEN) regulate several crucial cell functions such as proliferation, survival, genomic stability and cell motility through both phosphatidylinositol 3-kinase (PI3K)-dependent and -independent mechanisms. It reduces downstream AKT activity and, consequently, its partial or complete loss of function activates the PI3K-AKT pathway, causing a dysregulation in cell cycle control, apoptosis, cell contact and migration in different cancers [24].

Loss of function of the PTEN suppressor gene is not among the most frequent mutations in melanoma, being found in about 7–15% of cases [25]. However, it is usually coupled with BRAF-activating mutations and associated with reduced overall survival in BRAF-mutated patients [26,27,28]. Giles and colleagues demonstrated that analyzing PTEN epigenetic modifications could also help with identifying patients with an inactive gene and a worse prognosis [29]. PTEN mutations seems to occur later in the process of melanoma development and have been associated with resistance to immunotherapy and targeted therapy [29,30], reducing survival expectations in stage III melanoma [29]. However, the prognostic and predictive role of PTEN loss needs to be further elucidated in details, due to the exceptionally complex regulation of its functional status, which involves genetic, transcriptional, post-transcriptional and post-translational events.

### 2.5. MITF

Microphthalmia-associated transcription factor (MITF) is the key gene regulator of melanocyte development, differentiation, proliferation and survival. Its deregulation is involved in melanoma oncogenesis and, in particular, its amplification has been detected in approximately 15% of metastatic melanoma patients [31,32].

Disruption of MITF and its target CDKN2 has been shown to suppress growth and cell cycle progression in melanoma, but not in other cancers [33]. A recurrent activating point mutation in MITF has recently been discovered in cases of familial melanoma (above). Some experimental approaches using histone acetyltransferase inhibitors are ongoing [34].

In addition to its role as a potential therapeutic target, MITF is clinically useful for diagnostic purposes in identifying melanocytes at different evolutional stages and as an independent prognostic factor for predicting both malignancy and progression-free survival [35,36].

Furthermore, MITF is also a key determinant of melanoma cell plasticity and tumor heterogeneity in the tumor microenvironment, which are considered among major obstacles for effective immunotherapy [37].

### 2.6. CDKN2a

Cyclin-dependent kinase-2a and 2b (CDKN2a-CDKN2b) are located in the INK4 locus on chromosome 9p21 and encode respectively for tumor suppressor p16 and p15, which act as CDK inhibitors, being associated with a predisposition to melanoma [38]. Germline mutations or deletions resulting in p16 loss have been observed in up to 30% of melanomas and are associated with hereditary melanoma [39,40,41]. These alterations in the pathway of CDKN2A and Cyclin D1 (CCND1) are hypothesized to play a role in sensitivity to CDK inhibitors [42]. In particular, p16 inhibits CDK4 and CDK6, thereby preventing the formation of CDK/Cyclin D complexes that phosphorylate and activate the retinoblastoma protein. Loss of p16 contributes to tumorigenesis by favoring cell cycle progression [41,42]. Instead, the p14ARF protein inhibits p53 degradation by controlling MDM2 ubiquitin ligase to allow cell cycle arrest and apoptosis [43,44]. Partial or complete deletion of the INK4 gene cluster has been observed in most melanoma cell lines and in almost half of melanoma metastases [45,46]. Conway et al. found that reduced gene dosage of the regions of 9p21 encoding CDKN2A, CDKN2B and P14ARF was associated with increased tumor thickness, mitotic rate and ulceration [44]. Similarly, Grafström et al. reported that monoallelic or biallelic deletions in the INK4 region were associated with reduced median survival [46].

### 2.7. AXL

Receptor tyrosine kinase AXL is a transmembrane protein member of the TAM family, signaling through the growth of the arrest-specific protein 6 (GAS6/AXL) pathway, which influences tumor cell growth, metastasis, invasion, epithelial-mesenchymal transition (EMT), angiogenesis, drug resistance, immune regulation and stem cell maintenance [47,48]. When AXL is upregulated, it seems to impact on survival and therapy resistance in cancers, but also in neurofibromatosis type 1 and some inflammatory diseases [49].

Increasing soluble AXL (sAXL) levels in blood has been reported by Flem–Karlsen and colleagues as being associated with melanoma disease progression and correlated with shorter two-year survival in stage IV patients treated with ipilimumab [47]. Furthermore, sAXL levels were related to the percentage of cells expressing AXL in resected melanoma lymph node metastases [50]. Higher sAXL levels were also observed in late-stage melanoma patients compared to patients at an earlier stage, and sAXL levels were linked to a higher number of metastases and lower survival at week 7 of treatment [47].

Upregulation of the receptor tyrosine kinase AXL has been also linked with both a reduced response to immune checkpoint blockade as well as the development of therapy resistance to BRAF directed therapies in melanoma [51,52]. However, quantification of sAXL blood levels is a simple and easily assessable method to determine cellular AXL levels that could be further evaluated for use as a biomarker of disease progression and treatment response in melanoma [53]. Finally, a randomized phase Ib/II study of the selective small molecule AXL-inhibitor bemcentinib (BGB324) in combination with either dabrafenib/trametinib (D/T) or pembrolizumab in patients with metastatic melanoma is ongoing, exploring the efficacy and role of sAXL to monitor minimal residual disease [54].

### 2.8. Maspin

Maspin is a member of the serpin family of protease inhibitors involved in key processes of cancer progression. Its biological activity seems to be cancer and compartment- specific, with the protein acting either as a suppressor or as a tumor promoter in different cancer types. Maspin expression and its sub-cellular localization has been studied as a possible melanoma prognostic factor and related to melanoma progression [55]. In particular, Martinoli et al. evaluated nuclear and cytoplasmic maspin expression on 60 nevi, 152 primary lesions and 106 melanoma metastases, using tissue microarrays and immunohistochemistry. In univariate analysis, nuclear maspin expression in primary melanomas was significantly associated with aggressive phenotypes (nodular histotype, tumor thickness, mitotic rate and ulceration) and more advanced stages, whereas cytoplasmic maspin was observed more frequently in thin superficial spreading melanomas without mitosis. In multivariate analysis, nuclear maspin remained significantly associated with shorter disease-free and overall survival [55].

### 2.9. Tumor Mutational Burden (TMB)

The mix and number of somatic mutations in cancer cells constitutes the TMB, which is characterized by large variability across different solid tumor types. A high TMB is probably associated with a high neoantigen generation and, consequently, is considered a possible factor positively influencing immunotherapy effectiveness following the activation of the immune system [56,57,58].

In particular, lung cancers and melanomas are among the tumors with the highest TMB, which has been independently associated with better response rates (RR), PFS and overall survival (OS) in patients treated with anti PD1/PDL1 immunotherapy [59].

Interestingly, melanomas harboring NF1 or NRAS mutation are associated with higher TMB compared with other phenotypes [60], opening the opportunity to use immunotherapy with more solid scientific support.

## 3. Immunological Biomarkers

Impaired immunosurveillance and cancer-related inflammation are crucial events for cancer cell development and progression, since they influence tumor microenvironment composition (regulatory cells, myeloid derived cells, neutrophils) and T-lymphocytes activation through cytokine secretion. Furthermore, the expression of specific receptors and their ligands has been differently associated to patient outcomes. Qualitative and quantitative analysis of all those events could serve in improving patient selection for immunotherapy through the identification of specific biomarkers.

### 3.1. PD-1/PDL-1 Expression

PDL1 was widely investigated as a predictive biomarker of response to anti-PD1/PDL1 checkpoint-inhibitors. However, due to its highly discontinuous and dynamic expression, it is not particularly reliable. In fact, if a high PDL1 expression is constantly associated with better outcomes, a negative PDL1 status does not exclude a treatment response.

In patients with advanced melanoma enrolled in the Checkmate 037 study, the proportion of those achieving an objective response to second-line nivolumab was 44% when positive for PDL1 expression, and only 20% in those who were PDL1-negative [61]. Similar findings were reported in the Checkmate 066 trial using the combination ipilimumab/nivolumab [62]. In the KEYNOTE 001 study, which assessed pembrolizumab in 655 patients with advanced melanoma, PDL1 expression in pre-treatment tumor biopsy samples correlated with response rate, PFS and OS. However, some patients with tumors considered PD-L1-negative also achieved a durable response [63].

Although early studies pointed to PDL1 expression as a biomarker for increased sensitivity to PD1/PDL1-targeted agents, this relationship is not straightforward, probably due to its dynamic expression over time. The unpredictable nature of PDL1 as a biomarker has been demonstrated across different tumor types and with different PD1/PDL1-targeted antibodies. As a result, additional markers have been proposed and are emerging as potentially more useful predictive factors of response.

### 3.2. Tumour Infiltrating Lymphocytes

Immune and stromal cells within the tumor microenvironment are known to play an important role in influencing disease development as well as cellular response to therapies. Tumor infiltrating lymphocytes (TILs) appears to correlate with outcome, and increasing evidence suggests that their number, type and location within the primary tumor has prognostic value. In fact, a significantly higher density of CD8+ cells at both the invasive margin and the tumor center has been documented on pre-treatment samples from melanoma patients who benefited from anti-PD-1 treatment compared with those who experienced progression [64]. This has led to the development of the Immunoscore, a consensus scoring system that reflects the densities of CD3+ and CD8+ T-cell effectors present within the tumor and the invasive margins [65].

Immune cell infiltrate of colorectal cancer is one of the most well-studied score systems; however, the nature of tumor-infiltrating lymphocytes is heterogeneous and varies in different cancers. In advanced melanoma, the potential prognostic value of CD3, CD8, CD20 and FOXP3 expression levels is being evaluated in patients treated with ipilimumab [66], and the correlation of marker expression profile with clinical outcome is ongoing.

The Immunoscore system is emerging as a useful prognostic tool in different diseases, but it could also potentially help in predicting response to specific drugs and thus drive therapeutic decisions. Although this approach needs to be validated in order to become a more widely applicable and affordable combinatorial biomarker, measuring the quantity, quality and distribution of the immunologic tumor microenvironment has led to the development the concept of immunoprofiling, which means to characterize a specific tumor in order to personalize subsequent treatments [67].

### 3.3. Regulatory T Cells (Tregs) and Circulating Myeloid-Derived Cells (MDSCs)

Tregs and MDSCs are potential biomarker candidates, since they might limit the activity of checkpoint-inhibitor antibodies by promoting T cell exhaustion and dominant suppression in the tumor microenvironment.

In general, increased Tregs frequencies and a reduced CD8+/Tregs-ratio are linked to poor prognosis in multiple cancers [66]. In melanoma patients treated with ipilimumab, significantly lower levels of Lin^−^CD14^+^HLA^-^DR^−^ MDSCs have been observed in responders compared with non-responders [68]. Furthermore, another analysis of 209 advanced melanoma patients showed that baseline levels of low Lin^−^CD14^+^HLA^−^DR^−/^^low^-MDSCs and high CD4^+^CD25^+^FoxP3^+^-Tregs, were significantly associated with better survival [69].

### 3.4. CD-73

PD-1 signaling inhibition results in a sort of paradox: on one side, it induces the activation of CD8+ T cells and increases the production of effector molecules such as interferon-gamma (IFN-γ) and Granzyme Bl; on the other, it enhances the expression of adenosine receptor-2A, which in turn is able to limit CD8+ T cell response. In fact, tumor-derived CD-73 produces extracellular adenosine that suppresses anti-tumor immune response and promotes tumor growth via adenosine receptor signaling [70]. In addition, CD73 expression on tumor cells reduces the immune response evoked by anti-PD-1 and anti-CTLA-4 antibodies [71,72].

As a confirmation, adenosine receptor 2A blockade has been shown to increase the efficacy of anti-PD-1 therapy through enhanced antitumor T cell responses in mouse models [73]. Thus, CD73 expression on tumor cells has been suggested as a potential biomarker for response to anti-PD-1s.

### 3.5. IFN-γ Signature

The IFN-γ signaling pathway can result in resistance to immune checkpoint blockade [74]. IFN-γ can arrest tumor growth, augment MHC class I expression, increase recruitment of effector cells and coordinate innate and adaptive antitumor responses. However, IFN-γ signaling can also compromise anti-tumor immunity, inducing the expression of PD-L1 and up-regulating the expression of other immune-suppressive molecules in the tumor microenvironment [75]. In melanoma and lung cancer biopsies, high expression of IFN-γ has been associated with response and improved survival with anti PD-1 antibody treatment [76].

### 3.6. Tumour Inflammation Gene Signature

Multigene expression analysis of RNA levels can be used to characterize tumor and immune cells, indicating a T cell-inflamed phenotype which is considered related to enhanced clinical activity of anti-PD-1/PD-L1 agents. This extensive information can have prognostic and predictive relevance.

Tumor specimens from 19 melanoma patients treated with pembrolizumab within the KEYNOTE-001 study were analyzed by immune-related gene expression profile on the NanoString nCounter platform (NanoString Technologies Inc.) using a 680 tumor- and immune-related genes customized panel [74]. Genes associated with response were linked to IFN-γ signaling and showed a direct correlation with IFN-γ expression. Additionally, the panel identified other genes that showed a positive association with response and/or PFS, leading to a larger expanded 28-gene set referred to as the “preliminary expanded immune” signature. Moreover, an 18-gene NanoString T cell inflammation signature (TIS) that contains IFN-γ-responsive genes related to antigen presentation, chemokine expression, cytotoxic activity and adaptive immune resistance was also correlated with clinical benefit in patients treated with pembrolizumab [75], nivolumab and ipilimumab [77]. TIS was associated with response in patients treated with first-line anti-PD-1 (*p* = 0.012) and ipilimumab (*p* = 0.002, but not with response to anti-PD-1 agents as second-line treatment after ipilimumab and/or target-therapy in ongoing pembrolizumab trials.

It seems likely that the identification of further gene expression signatures that predict response to immunotherapy will be an increasing focus of research.

### 3.7. Lymphocyte Activation Gene-3 (LAG-3 or CD223)

LAG-3 is a checkpoint molecule expressed on activated T cells, natural killer cells, Tregs, Tr1 cells, exhausted T cells, B cells and dendritic cells and usually works in cooperation with other checkpoints, such as PD1/PDL1 and/or CTLA4 [78,79]. LAG-3 has been chosen as a cancer immunotherapeutic target because it is able to mediate a state of immune exhaustion. Synergy between LAG-3 and PD1s has been reported in tumor models, suggesting that dual immunotherapeutic inhibition would enhance efficacy and may extend to multiple tumor types [80].

Initial results with the anti-LAG-3 antibody relatlimab combined with nivolumab showed a nearly 3-fold higher response rate among patients with melanoma whose tumors expressed LAG-3 ≥ 1% versus those who had <1% LAG-3 expression (20% vs. 7.1%) [81]. Combination relatlimab/nivolumab in metastatic melanoma is under investigation in a phase III trial (NCT03470922). LAG-3 high expression appears to be associated with response and improved survival to checkpoints inhibition.

## 4. Peripheral Blood Markers

Systemic inflammation is associated with extended modifications in peripheral blood leukocytes composition, circulating B-lymphocytes, neutrophils, macrophages and mast cells. On the other hand, platelets release vascular endothelial growth factor (VEGF), which mediates leukocyte migration and extravasation, and platelet-derived growth factor (PDGF), which recruits neutrophils and monocytes.

All these cells and cytokines play an important role as effectors of innate immunity and initiators of adaptive immune responses. In this context, leucocyte absolute numbers and their differential counts/ratios can be used as a safe, easily assessable and cost-effective inflammatory index to identify subgroups of patients with different behavior during treatment.

Lactate dehydrogenase (LDH), C-reactive protein (CRP) and other cytokines, chemokines and secreted factors are also affordable but non-specific markers of disease progression and extension, being related to cell type and its functional orientation, tumor burden, disease localization, development of immune-mediated adverse events and prognosis.

### 4.1. NLR, PLR, MLR and Peripheral Blood Counts

Inflammation has been included among the hallmarks of cancer and recognized as an influencer for its development and progression. Elevated absolute neutrophils count (ANC), absolute lymphocyte counts (ALC), neutrophils/lymphocytes ratios (NLR) and derived-NLR (dNLR = ANC/WBC−ANC) are markers of systemic inflammation and independent predictors of survival in multiple cancer types. In melanoma, those advanced disease patients undergoing immunotherapies and showing elevated baseline ANC or NLR seem to have a poorer OS.

Gandini et al. analyzed data from 584 melanoma consecutive patients admitted in a comprehensive cancer center throughout a decade by using the institutional tumor registry [82]. In early-stage patients, peripheral blood cell counts were not associated either with the presence of active disease or with survival. Interestingly, when disease progressed from regional to distant sites, becoming metastatic, a significant increase of whole blood cells (WBC), and a change of peripheral blood cell composition were observed. These data were suggestive of an expansion of the myeloid compartment (neutrophils and monocytes) and a reduction of the lymphoid one. The same group demonstrated also that in stage IV patients, higher WBC, ANC, absolute monocyte counts (AMC), NLR and monocytes/lymphocytes ratio (MLR) were all directly associated with extent of disease and, independently of other prognostic factors, with increased risk of mortality.

The prognostic value of NLR in melanoma was also reported in a recent meta-analysis by Ding et al.: in a sample of 3207 patients, high NLR was associated with poor OS and PFS, independently from the choice of treatment [83].

Pre-therapy NLR was retrospectively analyzed by Ferrucci et al. in 69 metastatic melanoma patients treated with ipilimumab: patients with baseline NLR < 5 had a significantly improved PFS and OS compared with those with NLR ≥ 5. Similar findings were reported in three validation cohorts of patients receiving Ipilimumab in different settings [84] and by Valpione and colleagues [85]. dNLR and neutrophilia (ANC > 7500 cell/µL) were analyzed in an even larger cohort of melanoma patients (N = 720) receiving ipilimumab and included in the expanded access program across 51 Italian institutions [86]. Patients with baseline dNLR ≥ 3 had a 2.3-fold increased risk of death and a double-increased risk of disease progression when compared to patients with dNLR < 3 in multivariate analyses that included age, sex and LDH. Survival rates for patients with dNLR < 3 were 41% and 23% at 1 year and 2 years respectively, and 16% and 4% for patients with high dNLR. When the elevation of both dNLR and ANC was analyzed simultaneously, a 5.7-fold increased risk of death and a four-fold increased risk of progression was stated. As a consequence, ANC, dNLR and their combination could be considered as general prognostic, but not predictive, markers for response to ipilimumab.

Relative lymphocyte count (RLC) and relative eosinophil count (REC) were also evaluated by the same group in advanced melanoma patients receiving chemotherapy (n = 116) or anti-CTLA-4 therapy (n = 128) [87]. Authors observed that RLC ≥ 17.5% was associated with a significantly reduced risk of mortality in patients receiving either chemotherapy or anti-CTLA-4. In contrast, REC ≥ 1.5% had no prognostic value for patients treated with chemotherapy, while halving the risk of mortality in those receiving anti-CTLA-4, introducing an interesting discriminator marker between chemo- and immunotherapy.

Interestingly, Cocorocchio and colleagues showed that in patients receiving target-therapy, NLR ≥ 5 was significantly associated with increased risk of progression and mortality, either in univariate or in multivariate analysis, independently of age, sex, stage, LDH > 2xULN, previous treatments, concomitant use of steroids and type of therapy [88].

### 4.2. LDH and Multiparameter Combinations

In melanoma, LDH blood level is used as a general indicator of disease recurrence, with a sensitivity of 72% and specificity of 97%. Moreover, elevated LDH levels predict lower response rates to therapy and poorer survival, while monitoring LDH levels during treatment is complementary to normal imaging and able to provide information on response or disease progression. LDH was also the first marker identified as being associated with response to ipilimumab [89] and anti-PD-1 antibodies in monotherapy, or even in combination regimens [90], but not to duration of response [91,92].

Ferrucci et al. investigated the possibility of further combining ANC, dNLR and LDH in patients receiving ipilimumab. In multivariate analysis, the increase of these three parameters was associated to 13-fold increased risk of death compared to those who had none of these markers elevated. In addition, increased ANC, dNLR and LDH remained significantly associated with poor survival in presence of liver and/or brain metastasis [87].

A retrospective analysis of patients treated with pembrolizumab by Ribas showed that absence of visceral metastasis, high RLE ratio, high absolute lymphocyte count, good performance status, low LDH and low CRP are all independent prognostic factors for clinical outcome [92].

Multiparameter combinations, like the cancer immunogram, which includes tumor-specific and patient-specific markers to personalize evaluation, appeared to be superior to single biomarkers in order to predict response and direct treatment choices [93].

Similar studies have focused on possible markers able to identify the subset of metastatic melanoma patients who would benefit from targeted therapy. In particular, a pooled analysis of those treated with dabrafenib and trametinib showed LDH levels, number of metastatic organ sites and Eastern Cooperative Oncology Group (ECOG) performance status to be independent markers for deeper and durable responses [94,95].

### 4.3. Circulating Cytokines and Secreted Factors

Cytokines, chemokines, growth factors and other proteins are able to mediate interaction between peripheral immune systems and tumor microenvironments.

The cytokine profile can be used as a target for developing strategies of treatment, preventing toxicities and monitoring responses [96,97].

For example, IFN-γ up regulates tumor immune surveillance by decreasing production in lymphocytes, NK cells or PBMCs in many tumor types [98,99]. On the other hand, TNF-α, inducing immune tolerance through TGF-β [99,100], is associated with poor prognosis and metastatic behavior in several solid tumors. Indoleamine 2,3-dioxygenase (IDO), a tryptophan-degrading enzyme, is induced by inflammation and functions as an immunosuppressant with negative effects on proliferation, function and survival of T cells, also by promoting Treg development [100,101]. Other secreted molecules of interest are Granzymes, important for the ability of NK and CD8+ cells to kill target cells, and CXCL9/CXCL10, able to attract CD8+ cells enhancing MHC class I effector functions; both their expressions correlate with favorable prognosis in different cancers [102].

Measuring the cytokine milieu may offer clinical value, as cytokine profiles provide information on the functional orientation of cells and the local environment. Although assessing cytokine profiles in tissue biopsies remains a significant technical challenge, several platforms are available to assess cytokine profiles: ELISA, ELISPOT, intracellular cytokine staining, gene expression platforms and cytokine bead arrays [103]. Lim et al. examined the relationship between cytokine levels and patient response to immunotherapy on 65 cytokines. These cytokines were profiled longitudinally in 98 melanoma patients receiving mono-immunotherapy with anti-PD-1 and validated on 49 patients receiving combo immunotherapy with anti-PD-1 and anti-CTLA-4. Baseline expression of some cytokines was associated with treatment response and/or OS, but data were not constant amongst the cohorts and the findings warrant further investigation [104]. Less information is available on how circulating biomarkers may predict the risk of immune-related adverse events (irAEs). Lim et al. also report that plasma cytokine levels are generally stable at baseline or early during immune-modulating treatments, and that changing these parameters could reflect either the development of toxicities or have an impact on responses [104]. A toxicity score called CYTOX that includes the expression of 11 circulating cytokines has been developed and validated in patients treated with a combination PD-1 and CTLA-4-inhibitor. Elevated expression of these 11 cytokines (G-CSF, GM-CSF, Fractalkine, FGF-2, IFN-alfa2, IL-12p70, IL-1a, IL-1B, IL-1RA, IL-2, IL-13) was strongly associated with severe irAEs that required intervention with high-dose immunosuppressing agents.

### 4.4. Circulating Tumour DNA (ctDNA)

There is growing evidence on the biomarker role of ctDNA as an accurate predictor of tumor response, PFS and OS.

In particular, a recent metanalysis evaluated patients with metastatic melanoma and detectable ctDNA [105]. Detectable ctDNA at baseline and during follow-up was associated with poorer PFS and OS and appeared to be a strong prognostic biomarker for advanced-stage melanoma. In addition, ctDNA seems to be associated to baseline tumor burden (TB): metastatic melanoma patients with low baseline ctDNA and low TB have a wider response and longer PFS when treated with MAPK-inhibitors. Moreover, a reduction in ctDNA occurring during treatment is considered an independent predictor of response to BRAF inhibitor in melanoma [106].

Lee and colleagues examined whether ctDNA levels at baseline and early during therapy could predict response and clinical outcome in metastatic melanoma patients harboring BRAF, NRAS or CKIT mutations and receiving immunotherapy [107]. Even in this case, longitudinal assessment of ctDNA worked as an accurate predictor of tumor response, PFS and OS: patients who had a persistently elevated ctDNA on therapy had a poorer prognosis, possibly guiding differentiation of therapeutic choices.

Possible limitations on the use of ctDNA as a predictive biomarker are the low level of baseline ctDNA detectability and the absence of an identifiable driver mutation in up to 25% of melanoma patients.

## 5. Gut Microbiota Biomarkers

Growing evidence sustains bacterial microbiome as an important player in the early phase of carcinogenesis as well as in disease progression and in patient’s response to cancer treatments, with particular attention to immunotherapies [108,109,110,111,112,113].

Gopalakrishnan and colleagues examined the oral and gut microbiome of 112 melanoma patients undergoing anti-PD-1 immunotherapy [112]. When looking at the gut, but not oral, they observed significant differences in the diversity and composition of the microbiome of patients who responded to PD-1 therapy versus those patients who did not respond. Responders had higher alpha diversity (*p* < 0.01) and relative abundance of bacteria of the Ruminococcaceae family (*p* < 0.01) [110]. A similar study showed that patients who responded to anti- PD1 therapy had an abundance of Bifidobacterium longum, Collinsella aerofaciens and Enterococcus faecium compared with non-responders [113].

Wargo et al., on the other hand, demonstrated increased tumor immune infiltrates in responding patients, with higher density of CD8+ T cells correlating with abundance of specific bacteria in the gut microbiome [114].

Together, all these data suggest that increased bacterial diversity in some patients may lead to increased immune cell infiltration, and therefore, that the commensal microbiome may have a mechanistic influence on anti-tumor immunity.

Interestingly, the use of antibiotics and probiotics has been associated with reduced microbiome diversity and poorer response to immune checkpoints inhibitors (ICI) [115]. Instead, a diet rich in fiber was associated with five-fold higher odds of ICI response compared to a low-fiber one [116].

Another interesting observation is that the gut microbiome, aside from affecting response to therapy, may influence the development of adverse events, and be used as a biomarker to identify which patients are most at risk for experiencing collateral effects, like checkpoint blockade-induced colitis [117]. In fact, the pre-inflammation fecal microbiota and microbiome composition seems to be related to colitis development.

However, the intrinsic mechanisms on how microbiome may influence immunotherapy efficacy and toxicity remain unknown. Evidence for the immunomodulatory effect of the microbiome is building, and its translation to a potential therapeutic intervention is under investigation.

## 6. Conclusions

Since the treatment landscape for metastatic melanoma is continuously evolving, the availability of predictive/prognostic biomarkers able to select patients who are more likely or unlikely to benefit from a given therapy in advance of its initiation would allow for optimizing disease-management strategies. In addition, by giving patients the treatment they are best suited to receive, we could spare them unnecessary drug-related toxicities, improving their quality of life and reducing the costs of related support.

Unfortunately, the clinical utility of these markers for daily oncology practice is yet to be consolidated and, at present, not sufficiently reliable to guide treatment decisions.

Novel techniques may provide a more complete assessment of a tumor’s genomic landscape, augmenting sensitivity and specificity, though minimizing the impact of intratumoral heterogeneity.

In conclusion, the science around biomarkers is focusing on tumor and host genomics. Given the vast volume of data becoming available, significant investment in computational biology and artificial intelligence will be required to adequately weight and correlate all the different variables and their impact on clinical outcome. Moreover, all these processes must undergo rigorous standardization to maintain specificity and sensitivity while ensuring reproducibility, in order to demonstrate practical and financial viability to allow widespread application in the clinic (Table 1).

## Figures and Tables

**Table 1 cancers-14-00081-t001:** Prognostic, predictive and on-treatment biomarkers for advanced melanoma. General markers have been distinguished by Molecular, Cellular, Immunological and Humoral ones.

Prognostic Biomarkers	Predictive Biomarkers	On-Treatment Biomarkers
**General:** Stage (III vs. IV, IVa vs. IVc) Sites of metastasis Number of metastatic sites **Molecular:** BRAF V600 mutation NRAS mutation AXL **Cellular/immunological:** Multiparameter combos (NLR, PLR, MLR) Cancer immunogram Tregs and MDSCs T-cell Receptor (TCR) sequencing Gut microbiome **Humoral:** Lactate dehidrogenase (LDH) C-reactive protein Circulating tumor DNA	**Molecular:** BRAF V600, CKIT mutations Tumor Mutational Burden (TMB) PTEN loss of function CDKN2a sAXL **Immunological:** Checkpoints expression (PD-1/PDL-1, LAG-3) Immunoscore (CD8+ T-cell infiltration) CD-73 expression IFN-gamma signature Tumor inflammation signature T-cell Receptor (TCR) sequencing Gut microbiome **Humoral:** Cytox score T-cell receptor (TCR) sequencing Circulating tumor DNA	**Molecular:** Circulating tumor DNA sAXL **Cellular/immunological:** Absolute lymphocyte count **Proliferating** CD8+ T-Cells T-cell subsets increase (CD8+, TREGS) Granzyme B expression **Humoral:** Cytox score

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
