# Peer review of "Novel Biomarkers and Druggable Targets in Advanced Melanoma"

_cancers, 2021, doi:10.3390/cancers14010081_

Round 1

Reviewer 1 Report

In this review article, the authors discuss the roles of various biomarkers in indicating possible responses to treatment or development of toxicities in advanced melanoma patients. This paper covers various aspects, and may be informative for readers.

Several revisions are required.

1) Line 102: “increased” should be “increase”.

2) Line 177: “It not” should be “it is not”.

3) Line 300: Full spelling is required for “PLR”.

4) Line 304: Explanation is required for “derived-NLR”.

Author Response

Reviewer 1 several requested revisions are aswered:

1) Line 102: “increased” has been changed in “increase”.

2) Line 177: “It not” has been changed in “it is not”.

3) Line 300: Full spelling has been added for “PLR”.

4) Line 304: Explanation formula has been added for “derived-NLR”.

Reviewer 2 Report

This is a reasonable review of prognostic and predictive biomarkers in Melanoma. Some editorial revisions need to be made before it can be accepted. 

1) There are numerous improper language issues, with some listed below. I am sure there are others that can be caught and corrected through careful editing.

Line 13: "easily to detect" is not correct;

Line 69: "lead" is not the right word;

Line 149: "lately" should be late;

Line 151: "at moment" is not proper;

Line 213: "wider" should be widely;

Line 215: "the development the concept" is incorrect;

Line 384: "in several solid." seems to be missing something.

2) In section 4.2, it should be made clear it is the blood LDH level.

3) The abbreviations in the subtitles should be spelled out, especially when they appear before they are defined in the text.

4) Some sections contain multiple one-sentence paragraphs, such as 4.3, and 6. They are fragmented and should be revised.

5) why is "baseline tumor burden" abbreviated as TM?

Author Response

As kindly requested by reviewer 2:

-All the numerous improper language issues,  have been addressed.

-In section 4.2, it has been made clear that is the blood LDH level.

-Abbreviations has been controlled and spelled out before they are defined in the text.

- One-sentence paragraphs, if intended as long and unclear sentences such as 4.3, and 6 have been revised.

- "baseline tumor burden" has been coorectly abbreviated as baseline TB?

Reviewer 3 Report

The review manuscript Novel Biomarkers and Druggable Targets in Advanced Melanoma (by Pier Francesco Ferrucci, and Emilia Cocorocchio) deals with some molecular targets suitable for targeted therapy including immunological aspects of treatment. Stress is given to immunological markers with only brief mention about other genes/markers (above). Thus, the MS seems rather like the „immunocharacterization“ or „immunomarkers“ in melanoma.

Major concerns:

Lines 144, 153 and 159: Some of the marker/target molecules are mentioned and commented only very briefly (eg. MITF, PTEN or CDKN2a), with only very limited references. Eg. MITF is an important and very specific melanoma marker that should deserve more description and mentioning its main function in melanoma cells and appropriate more and suitable references. PTEN should be also described in a slightly more detail.

Stress is given to immunological markers with only brief mention about other genes/markers (above). Thus, the MS seems rather like the „immunocharacterization“ or „immunomarkers“ in melanoma. Also, blood markers, mostly inflamatory and immunological, are discussed in a greater detail. When compares to driver mutation gene markers, the MS seems rather unbalanced.

Line 159:    sub-chapter 2.7. CDKN2a is also two short with also one citation. Eg. p16 was a long time ago considered as a “melanoma gene”. In my opinion, if “biomarkers” is in the title, more stress should be given to these crucial genes/markers in melanoma.

Conclusively, the MS is good written and overal gives some valuable information. The „oncogenic“ genes/markers are however rather underestimated and just shortly described. This could be improved by either: a) mentioning the „drivers“ and oncogenes (some of which are event missing-eg. Axl kinase and others) in a much broader manner, adding references and thus balance the article., or b) just mention the „drivers“ briefly in the introduction and continue with immunological or immunomodulating molecules, blood cell markers and other markers already described in the text. In the second case, the slight appropriate amendment of the title would be useful.

Minor:

Line 69: … „lead“ might be replaced with a more suitable word

Line 84: A notorious resistence and recurrence of BRAFV600E positive tumors after BRAF-i treatment as monotherapy should be stressed, with appropriate reference(s).

Line 88:  ..mechanisms..

Line 99: The reasons of BRAF-i resistence are numerous and multiple, sometimes molecularly very distant (eg. Life 11, 2021)

Line 132: 2.4. Tumor Mutational Burden (TMB) : this sub-chapter should be rather broader and placed as a last chapter in the section.

Line 145:    …melanomas…

Line 169: while the presence in the tumour microenvironment of T-cells, regulatory cells, myeloid derived cells and neutrophils or the presence of specific receptors and their ligands, are differently associated to the outcome.    -----  Please reorganize the sentence to be grammatically correct.

Author Response

We agree that molecular biomarkers should deserve more description and refences, so, thanks to reviewer suggestion, we extended that section adding information on CDK2 a/b, MITF, PTEN and other potential novel biomarkers.

The manuscript appear now much more balanced as requested.

All minor correction had been addressed, apart for line 84 where resistance to BRAF inhibitors is mentioned. In this case we think that resistance is out of the focus of our review and could be redundant information.

Reviewer 4 Report

In this manuscript Ferrucci and Cocorocchio reviewed both estabilished and innovative biomarkers and druggable targets belonging to 4 different categories (Molecular, Immunological, Peripheral and Gut/Microbioma) in order to evaluate their prognostic/predictive/on treatment rolein the management of melanoma patients.

The review is well organised, clear and give a comprehensive description of the state of art in this field.

A minor revision of English language, including punctuation, is needed, as there are minor language mistakes in the text [eg. pp. 3, line 101; pp.4, line 177; pp.5 line 200 etc].

Author Response

As requested by reviewer 4, English language has been controlled in the whole document by a mother tongue specialist and mistakes corrected.

Round 2

Reviewer 3 Report

The revised review manuscript Novel Biomarkers and Druggable Targets in Advanced Melanoma (by Pier Francesco Ferrucci, and Emilia Cocorocchio) deals with some molecular targets suitable for targeted therapy including immunological aspects of treatment.

Factual errors should be avoided, as suggested below.

Minor concerns:

Line 107:   ..provide…

Line 128: In several phase II trials…….Imanitib….KIT……    However, …….effectiveness.

--------- Please add a reference to this statement

Just a comment: other more effective (with lower IC50) inhibitors are now available for inhibiting KIT (e.g. masitinib (AB1010, probably? also used in clinical trials).

Line 173: Cyclin-dependent kinase-2A and 2B (….

 --------should be CDK inhibitor (CDKN2a codes for p16 tu. suppressor and CDK inhibitor). CDKN2b codes for p15 protein-also CDK inhibitor. ARF is not a CDK inhibitor-it inhibits p53 degradation (correctly given below in text) by controlling MDM2 ubiquitin ligase. 

-------please correct

Author Response

Thank you for the suggestions which have been all addressed in the text.